# Searching towards Class-Aware Generators for Conditional Generative Adversarial Networks

## Abstract

Conditional Generative Adversarial Networks (cGAN) were designed to generate images based on the provided conditions, *e.g*., class-level distributions. However, existing methods have used the same generating architecture for all classes. This paper presents a novel idea that adopts NAS to find a distinct architecture for each class. The search space contains regular and class-modulated convolutions, where the latter is designed to introduce class-specific information while avoiding the reduction of training data for each class generator. The search algorithm follows a weight-sharing pipeline with mixed-architecture optimization so that the search cost does not grow with the number of classes. To learn the sampling policy, a Markov decision process is embedded into the search algorithm and a moving average is applied for better stability. We evaluate our approach on CIFAR10 and CIFAR100. Besides achieving better image generation quality in terms of FID scores, we discover several insights that are helpful in designing cGAN models.

## 1 Introduction

Generative Adversarial Network (GAN) (Goodfellow et al., 2014) has attracted considerable attention and achieved great success in image generation. Conditional GAN (cGAN) (Mirza & Osindero, 2014) is a type of GAN using class information to guide the training of the discriminator and generator so that it usually obtains a better generation effect. Most cGANs incorporate class information into the generator through Conditional Batch Normalization (*CBN*) (de Vries et al., 2017), or into the discriminator through projection discriminator (Miyato & Koyama, 2018), multi-hinge loss (Kavalerov et al., 2019), auxiliary loss (Odena et al., 2017), *etc*.

In this paper, we investigate the possibility of designing class-aware generators for cGAN (*i.e*., using a distinct generator network architecture for each class). To automatically design class-aware generators, we propose a neural architecture search (NAS) algorithm on top of reinforcement learning so that the generator architecture of each class is automatically designed. However, as the number of classes increases, there are three main issues we have to consider. First, the search space will grow exponentially as the number of categories grows (*i.e*., combinatorial explosion). Second, training the generator separately for each class is prone to insufficient data (Karras et al., 2020). Furthermore, searching and re-training each generator one by one may be impractical when the number of generators is large.

We propose solutions for these challenges. First, we present a carefully designed search space that is both flexible and safe. We refer to **flexibility** as the ability to assign a distinct generator architecture to each class, which makes the search space exponentially large while its size is still controllable. To guarantee the **safety** (*i.e*., enable the limited amount of training data to be shared among a large number of generators), we introduce a new operator named Class-Modulated convolution (*CMconv*). *CMconv* shares the same set of convolutional weights with a regular convolution but is equipped with a standalone set of weights to modulate the convolutional weights, allowing the training data to be shared among different architectures and thus alleviating the inefficiency on training data. Second, to make the procedure of search and re-training as simple as possible, we develop *mixed-architecture optimization*, such that the training procedure of multiple class-aware generators is as simple as that of training only one generator.

Integrating these modules produces the proposed Multi-Net NAS (MN-NAS). To the best of our knowledge, this is the first method that can produce a number of generator architectures, one for

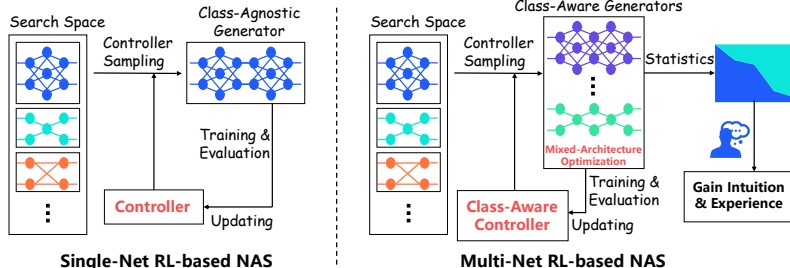

Figure 1: Illustration of Single-Net RL-based NAS and our Multi-Net RL-based NAS. Class-aware generators allow us to discover statistical laws by analyzing multiple network architectures. However, this cannot be achieved by class-agnostic generator because only one generator is searched (*i.e.*, one sample), so we cannot get more information for architecture design.

each class, through one search procedure. Figure 1 shows the overall framework of MN-NAS. It applies a Markov decision process equipped with moving average as the top-level logic for sampling and evaluating candidate architectures. After the search procedure, the optimal architecture for each class is determined and they get re-trained and calibrated for better image generation performance.

We perform experiments on some popular benchmarks, including the CIFAR10 and CIFAR100 datasets that have different numbers of classes. We achieve FID scores of $5.85$ and $12.28$ on CIFAR10 and CIFAR100 respectively, which are comparable to state-of-the-art results. In addition to achieving good performance, our method has given us some inspiration. For example, we find the phenomenon that the coordination between the discriminator and generator is very important (*i.e.*, to derive distinct class-aware generators, the discriminator must also be class-aware). More interestingly, by analyzing the best model found by NAS, we find that the class-modulated convolution is more likely to appear in the early stage (close to the input noise) of the generator. We think this phenomenon is related to the semantic hierarchy of GANs (Bau et al., 2018; Yang et al., 2020). We apply this finding as an empirical rule to BigGAN (Brock et al., 2018), and also observe performance gain. This implies that our algorithm delivers useful and generalized insights to the design of cGAN models. We will release code and pre-trained models to facilitate future research.

## 2   RELATED WORK

Generative Adversarial Network (GAN) (Goodfellow et al., 2014) have demonstrated impressive generation capabilities (Karras et al., 2017; Brock et al., 2018; Karras et al., 2019a). Nevertheless, it has notorious issues like vanishing gradient, training instability, and mode collapse. There are a number of improvements for the original GAN, *e.g.*, changing the objective function (Arjovsky et al., 2017; Gulrajani et al., 2017; Mao et al., 2016; Jolicoeur-Martineau, 2019; Qi, 2017), improving network architecture (Radford et al., 2015; Brock et al., 2018; Karras et al., 2019a; Denton et al., 2015; Zhang et al., 2018; Karnewar & Wang, 2019), using multiple generators or discriminators (Tolstikhin et al., 2017; Hoang et al., 2018; Arora et al., 2017; Durugkar et al., 2017; Ghosh et al., 2018; Nguyen et al., 2017). Recently, the surge in neural architecture search (NAS) has triggered a wave of interest in automatically designing the network architecture of GAN (Wang & Huan, 2019; Gong et al., 2019; Tian et al., 2020b; Gao et al., 2019; Tian et al., 2020a; Li et al., 2020; Fu et al., 2020; Kobayashi & Nagao, 2020).

Conditional GAN (cGAN) (Mirza & Osindero, 2014) is another type of GAN that incorporates class information into the original GAN, so that achieving promising results for the class-sensitive image generation task. Most of the early methods just incorporated the class information by concatenation (Mirza & Osindero, 2014; Reed et al., 2016). AC-GAN (Odena et al., 2017) incorporated the label information into the objective function of the discriminator by an auxiliary classifier. Miyato & Koyama (2018) proposed the class-projection (*cproj*) discriminator, which injected class information into the discriminator in a projection-based way. Furthermore, conditional batch normalization (*CBN*) (de Vries et al., 2017) is a very effective method to modulate convolutional feature maps by conditional information. Subsequently, *cproj* and *CBN* are widely used together, forming some powerful cGANs for class image generation (Zhang et al., 2018; Brock et al., 2018).

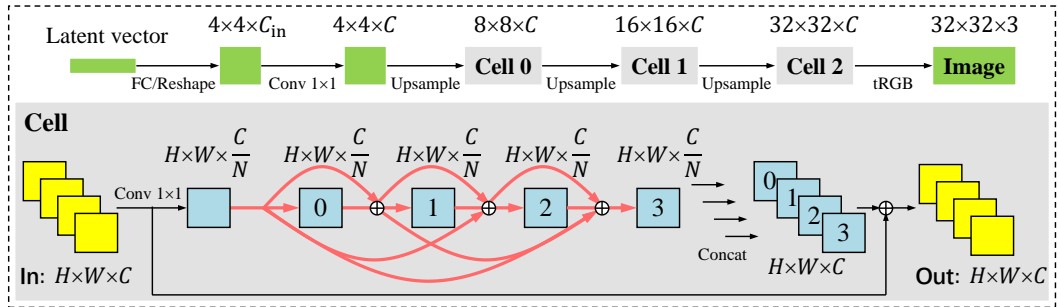

Figure 2: A tentative architecture in the search space. $N$ stands for the number of nodes in a cell, set to $4$ as an example. The operators shown in red arrows can be searched. The data shape is unchanged within each cell, so that the number of cells can be arbitrary. This figure is best viewed in color.

# 3 OUR APPROACH

We use NAS to design class-aware generators for cGAN. However, implementing this cGAN model is not a trivial task. First, it is not easy to define the search space and the generator of each class may suffer insufficient training data. To tackle these issues, we detail the search space and a weight sharing strategy in Section 3.1. Second, we must design an efficient search method. In Section 3.2, we introduce the Multi-Net NAS (MN-NAS) and the *mixed-architecture optimization*, these methods making the procedure of search and re-training of multiple networks simple.

## 3.1 SEARCH SPACE: SHARING DATA BY CLASS-MODULATED CONVOLUTION

The design of the search space follows the popular cell-based style. The input latent vector is passed through $L$ up-sampling layers, each of which is followed by a cell that does not change the shape (spatial resolution and channel number) of the data. Each contains an input node, an output node, and $N$ intermediate nodes and there exists an edge between each pair of nodes, propagating neural responses from the lower-indexed node to the higher-indexed node. Each node summarizes inputs from its precedents, *i.e.*, $\mathbf{x}_j = \sum_{i<j} o_{i,j} (\mathbf{x}_i)$ where $o_{i,j} (\cdot)$ is the operator on edge $(i, j)$, chosen from the set of candidate operators, $\mathcal{O}$. To guarantee that the shape of data is unchanged, at the beginning of each cell, the data is pre-processed using a $1 \times 1$ convolutional layer that shrinks the number of channels by a factor of $N$. Hence, the output of intermediate nodes, after being concatenated, recover the original data shape. An architecture with tentative operators is shown in Figure 2.

Since the operator used in each edge can be searched, the number of different architectures is $|\mathcal{O}|^{L \times \binom{N+1}{2}}$. Note that we allow each class to have a distinct architecture, therefore, if there are $M$ classes in the dataset, the total number of possible combinations is $|\mathcal{O}|^{L \times \binom{N+1}{2} \times M}$. This is quite a large number. Even with the simplest setting used in this paper (*i.e.*, $|\mathcal{O}| = 2$, $L = 3$, $N = 2$), this number is $2^{90} \approx 1.2 \times 10^{27}$ for a 10-class dataset (*e.g.*, CIFAR10) or $2^{900} \approx 8.5 \times 10^{270}$ for a 100-class dataset (*e.g.*, CIFAR100), much larger than some popular cell-based search spaces (*e.g.*, the DARTS space (Liu et al., 2019; Chen et al., 2019; Xu et al., 2020) with $1.1 \times 10^{18}$ architectures).

**Class-Modulated Convolution.** The first challenge we encounter is that the training data need to be distributed among all classes. That being said, if the architectures of all classes are 'truly' independent (*e.g.*, all network weights are individually optimized), the amount of training data for each architecture, compared to the scenario that all classes share the same architecture, is reduced by a factor of $M$. This can result in severe over-fitting and unsatisfying performance of image generation. To alleviate this risk, we share training data among different architectures by reusing model weights. In practice, most network weights are contributed by the convolutional layers, so we maintain one set of convolutional kernels and use a light-weighted module to introduce class-conditional information. Inspired by *CBN* (de Vries et al., 2017) and the 'demodulation' operation (Karras et al., 2019b), we propose the Class-Modulated convolution (*CMconv*) operator to incorporate class-conditional information. As shown in Figure 3, a *CMconv* layer consists of three parts, *modulation*,

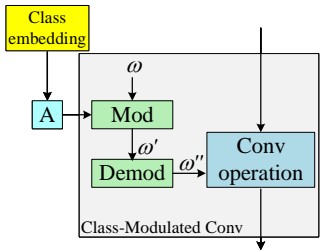

Figure 3: A class-modulated convolution (*CMconv*), where the class-conditional vector (class embedding) is used to modulate the convolutional weights (shared with the regular convolution).

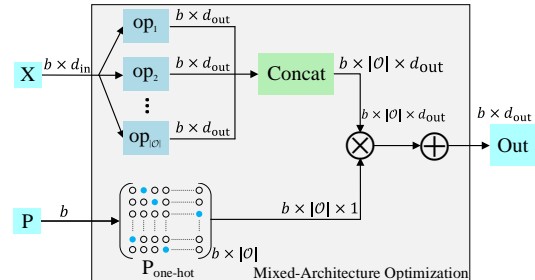

Figure 4: The mixed-architecture optimization with parallelization in a mini-batch. $\otimes$ stands for tensor-broadcasting multiplication, and $\oplus$ represents sum along the second dimension of the input .

*demodulation*, and *convolution*. The *CMconv* shares convolutional weights with the corresponding regular convolution (*Rconv*).

Mathematically, let $\mathbf{x}$ denote the input features maps with a class label of $y$, and $\boldsymbol{\omega}$ represent the weights of convolution. The goal of *modulation* is to introduce a scale factor to each of the input channels, *i.e.*, $\boldsymbol{\omega}' = \boldsymbol{\omega} \odot \mathbf{s}_{\text{in}}$ where both $\boldsymbol{\omega}$ and $\boldsymbol{\omega}'$ are in a shape of $c_{\text{in}} \times c_{\text{out}} \times U$. Here, $c_{\text{in}}$ and $c_{\text{out}}$ are the number of input and output channels, respectively, and $U$ is the kernel size (*e.g.*, $3 \times 3$); $\mathbf{s}_{\text{in}}$ is a $c_{\text{in}}$-dimensional vector and $\boldsymbol{\omega} \odot \mathbf{s}_{\text{in}}$ multiplies each set of weights ($c_{\text{out}} \times U$ numbers) by the corresponding entry in $\mathbf{s}_{\text{in}}$. We follow the conventional formulation of cGAN to define $\mathbf{s}_{\text{in}}$ as an affine-transformed class vector, *i.e.*, $\mathbf{s}_{\text{in}} = \text{Aff}(\mathbf{e}_y)$, where $\text{Aff}(\cdot)$ is simply implemented as a trainable fully-connected layer and $\mathbf{e}_y$ is a learnable embedding vector of the class label, $y$. The goal of *demodulation* is to normalize the weights and keep the variance of the input and output features same. We follow Karras et al. (2019b) to use $\boldsymbol{\omega}'' = \boldsymbol{\omega}' \odot \mathbf{s}_{\text{out}}^{-1}$ where $\mathbf{s}_{\text{out}}$ is a $c_{\text{out}}$-dimensional vector and $\mathbf{s}_{\text{out}}^{-1}$ indicates element-wise reciprocal; $\mathbf{s}_{\text{out}}$ is similar to the $\ell_2$-norm of $\boldsymbol{\omega}'$, computed as $\mathbf{s}_{\text{out}} = \sqrt{\sum_{c_{\text{in}},u}(\omega'_{c_{\text{in}},\cdot,u})^2 + \epsilon}$, where $\epsilon$ is a small constant to avoid numerical instability.

In summary, *Rconv* and *CMconv* start with the same weight, $\boldsymbol{\omega}$, and *CMconv* modulates $\boldsymbol{\omega}$ into $\boldsymbol{\omega}''$. Then, regular convolution is performed, *i.e.*, $\text{conv}(\mathbf{x}; \boldsymbol{\omega})$ or $\text{conv}(\mathbf{x}; \boldsymbol{\omega}'')$. Since *modulation* and *demodulation* introduce relatively fewer parameters compared to convolution, so using weight-sharing *Rconv* and *CMconv* operators in each edge is a safe option that enables the limited amount of training data to be shared among a large number of generators.

### 3.2 SEARCH METHOD: MULTI-NET NAS

The search process is formulated by a one-step Markov Decision Process (MDP). We denote $a$ as the action that samples architectures for all classes. Let $\pi(a; \boldsymbol{\theta}) \in (0, 1)$ be the sampling policy and $\boldsymbol{\theta}$ the learnable parameters, the performance of $\pi(a; \boldsymbol{\theta})$ is measured by:

$$J(\boldsymbol{\theta}) = \mathbb{E}_{a \sim \pi(a; \boldsymbol{\theta})}[R(a)], \tag{1}$$

where $R(a)$ is the reward function. Throughout this paper, we use the Inception Score as the reward. According to REINFORCE (Williams, 1992), the gradient of $J(\boldsymbol{\theta})$ with respect to $\boldsymbol{\theta}$ can be computed as:

$$\nabla_{\boldsymbol{\theta}} J(\boldsymbol{\theta}) = \mathbb{E}_{a \sim \pi(a; \boldsymbol{\theta})}[(R(a) - r) \cdot \nabla_{\boldsymbol{\theta}} \log(\pi(a; \boldsymbol{\theta}))] \approx \frac{1}{m} \sum_{k=1}^{m} (R(a_k) - r) \cdot \nabla_{\boldsymbol{\theta}} \log(\pi(a_k; \boldsymbol{\theta})), \tag{2}$$

where $m$ is the number of sampled architectures and $r$ is a baseline reward, set to be the moving average and used to reduce the variance in the training process. We use gradient ascent to maximize $J(\boldsymbol{\theta})$. Inspired by Cai et al. (2019); Ying et al. (2019), we design a simple policy. We use $\boldsymbol{\theta}_{l,k}$ to denote a $|\mathcal{O}|$-dimensional parameter of class $k$ and layer $l$, so that the probability of sampling

each operator, $\text{Prob}\left(o|\boldsymbol{\theta}_{l,k}\right)$, is determined by the softmax output of $\boldsymbol{\theta}_{l,k}$. Hence, given class-aware architectures, $A_{\text{ca}}$, the probability that it gets sampled is $\text{Prob}\left(A_{\text{ca}}|\pi\left(a;\boldsymbol{\theta}\right)\right)=\prod_{l,k}\text{Prob}\left(o|\boldsymbol{\theta}_{l,k}\right)$.

### 3.2.1 Overall Pipeline: Search, Re-Training, and Calibration

There are two sets of parameters to be learned, *i.e.*, $\boldsymbol{\theta}$ for the sampling policy and $\boldsymbol{\omega}$ for the super-network. We use the hinge adversarial loss (Lim & Ye, 2017):

$$\begin{aligned}
\mathcal{L}_D &= \mathbb{E}_{q(y)}\left[\mathbb{E}_{q(\mathbf{x}|y)}\left[\max\{0,1-D\left(\mathbf{x},y\right)\}\right]\right]+\mathbb{E}_{q(y)}\left[\mathbb{E}_{p(\mathbf{z})}\left[\max\{0,1+D\left(G\left(\mathbf{z},y\right),y\right)\}\right]\right], \\
\mathcal{L}_G &= -\mathbb{E}_{q(y)}\left[\mathbb{E}_{p(\mathbf{z})}\left[D\left(G\left(\mathbf{z},y\right),y\right)\right]\right],
\end{aligned}$$
(3)

where $\mathbf{x}$ with class label $y$ is sampled from the real dataset, $D\left(\cdot\right)$ and $G\left(\cdot\right)$ denote the discriminator and generator, respectively, and $p\left(\mathbf{z}\right)$ is the standard Gaussian distribution. We use the discriminator in AutoGAN (Gong et al., 2019) and add class projection (*cproj* (Miyato & Koyama, 2018)) to it. We emphasize that using a class-aware discriminator is critical to our algorithm (please refer to Section 4.2.1).

During the **search** phase, we adopt the weight-sharing NAS approach (Pham et al., 2018) to optimize the generator, *i.e.*, the super-network parameterized by $\boldsymbol{\omega}$. We perform fair sampling strategy (Chu et al., 2019) to offer equal opportunity for training each generator architecture. After every $T_{\text{critic}}=5$ iterations, we update the generator weights; after every $T_{\text{policy}}=50$ iterations, we update the policy parameters, $\boldsymbol{\theta}$. The pseudo code of the optimization process is provided in Appendix A.

After the search, we obtain the generator architecture for each class by choosing the operator with the largest score on each edge, *i.e.*, $o_{l,k}=\arg\max_o\text{Prob}\left(o|\boldsymbol{\theta}_{l,k}\right)$. Then, we **re-train** the generator from scratch following the same procedure. The last step is named **calibration**, in which we fine-tune the each architecture on the corresponding class for a small number of iterations (thus the overhead is small). As we will show in experiments, the calibration step largely boosts the performance, because the re-training stage has pursued for the optimality over all classes, which does not necessarily align with the optimality on each individual class.

### 3.2.2 Sharing Computation by Mixed-Architecture Optimization

We notice a technical issue that greatly downgrades the efficiency of both search and re-training. Given a mini-batch, $\mathcal{B}$, from the training set, as $\mathcal{B}$ may contain features from multiple classes, they need to be propagated through different architectures. To avoid heavy computational burden[1], we propose *mixed-architecture optimization* to allow different architectures to be optimized within one forward-then-backward pass in a batch.

The flowchart of mixed-architecture optimization is shown in Figure 4. Let $\mathbf{X}$ denote a batch of input features of size $b\times d_{\text{in}}$, where $b$ is the size of the batch, $\mathcal{B}$, and $d_{\text{in}}$ is the feature dimensionality. To improve the efficiency of parallelization, $\mathbf{X}$ as a whole is propagated through every operator, and each training sample chooses the output corresponding to the selected operator. In practice, this is implemented by concatenating all the outputs of $o\left(\mathbf{X}\right)$, $o\in\mathcal{O}$, and multiply it by $\mathbf{P}$, a $b\times|\mathcal{O}|$ indicator matrix. Each row of which is a one-hot, $|\mathcal{O}|$-dimensional vector indicating the operator selected by the corresponding sample.

Essentially, mixed-architecture optimization performs redundant computation to achieve more efficient parallelization. Each input feature is fed into all $|\mathcal{O}|$ operators, though only one of the outputs will be used. Nevertheless, $b\times|\mathcal{O}|$ does not increase with the number of classes and thus our method generalizes well to complex datasets, *e.g.*, CIFAR100.

## 4 Experiments

**Dataset and Evaluation.** We use CIFAR10 and CIFAR100 (Krizhevsky et al., 2009) as the testbeds. Both datasets have 50,000 training and 10,000 testing images, uniformly distributed over

---

[1]If we deal with each class in a batch individually, the corresponding architectures need to be loaded to the GPU one by one and the batch size becomes small, which results in a reduction in computational efficiency. Moreover, inefficiency deteriorates with the number of classes increases.

Table 1: FID scores of class-agnostic and class-aware GANs on CIFAR10.

| Method | Intra FIDs ↓ | | | | | | | | | | FID ↓ |
| --- | --- | --- | --- | --- | --- | --- | --- | --- | --- | --- | --- |
| | airp. | auto. | bird | cat | deer | dog | frog | horse | ship | truck | |
| NAS-cGAN | 29.10 | 13.62 | 26.64 | 22.21 | 14.97 | 26.02 | 17.32 | 15.18 | 14.99 | 16.58 | 7.05 |
| NAS-cGAN (calibrated) | 26.83 | 13.05 | 25.99 | 21.24 | 14.56 | 23.39 | 17.32 | 15.17 | 14.46 | 14.99 | 6.63 |
| NAS-caGAN | 29.53 | 12.32 | 24.83 | 21.30 | 16.11 | 26.64 | 16.56 | 16.79 | 16.43 | 16.39 | 6.83 |
| NAS-caGAN (calibrated) | 25.36 | **11.91** | **22.66** | **19.63** | **13.74** | 23.29 | **15.81** | 15.60 | 13.82 | **14.78** | **5.85** |
| NAS-caGAN-light | 35.21 | 12.64 | 29.04 | 24.96 | 21.20 | 26.26 | 18.62 | 16.40 | 14.87 | 18.67 | 7.79 |
| NAS-caGAN-light (calibrated) | **23.90** | 12.69 | 24.13 | 23.36 | 18.39 | **22.12** | 16.93 | **14.71** | **13.17** | 15.01 | 6.31 |

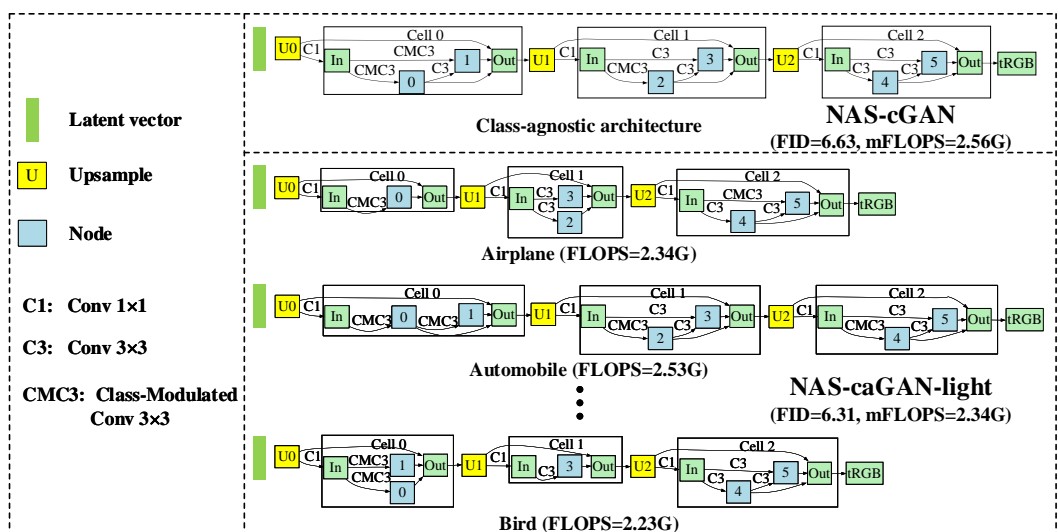

Figure 5: Part of the generator architectures found by NAS-caGAN-light on the CIFAR10 dataset. Compared to the class-agnostic architecture (shown at the top), the class-aware architectures enjoy a lower FID score as well as cheaper computation. This figure is best viewed in color.

10 or 100 classes. We use the Inception Score (IS) (Salimans et al., 2016) and the Fréchet Inception Distance (FID) (Heusel et al., 2017) to measure the performance of GAN on 50K randomly generated images. The FID statistic files are pre-calculated using all training images. We also compute the FID score within each class of CIFAR10. Specifically, for each class, we first use 5K real training images to pre-calculate the statistic file, and then randomly generate another 5K images for computing the intra FID score. For more experimental details, please refer to Appendix B.

## 4.1 QUANTITATIVE RESULTS

Table 1 summarizes the image generation results on CIFAR10. We use NAS-cGAN to denote the method that each class uses the same searched generator architecture, and NAS-caGAN to denote the method with class-aware generators incorporated. The former option is achieved by a modified version of the proposed search algorithm that the sampling policy is shared among all classes. It can be seen that NAS-caGAN produces lower FID scores, indicating its better performance compared to NAS-cGAN. After calibration, NAS-caGAN achieves even better results, reporting an FID of 5.85. We also compare the calibrated versions of NAS-cGAN and NAS-caGAN and find that the latter is better, indicating that both class-aware architectures and calibration contribute to the generating better images.

We notice that the current space chooses each operator between *RConv* and *CMConv*, both of which are parameterized and expensive. To find computationally efficient architectures, we add the *zero* operator into the search space, and without any further modification, derive a light-weighted version of class-aware generators, denoted as NAS-caGAN-light. With or without calibration, NAS-caGAN-light achieves comparable FID values, while enjoying a reduced average computational overhead (2.34G FLOPs vs. 2.56G FLOPs) of NAS-caGAN. This is another merit of using class-aware ar-

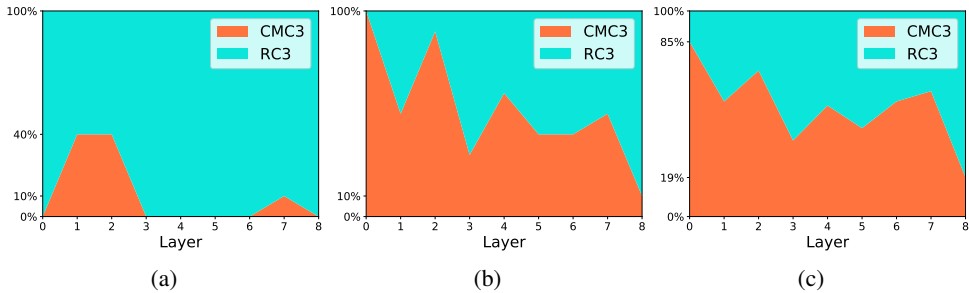

Figure 6: The proportion of *RConv* and *CMConv* in each layer of the searched class-aware architectures, where (a) is obtained using a normal discriminator on CIFAR10, (b) is obtained using a *cproj* discriminator on CIFAR10, and (c) is obtained using a *cproj* discriminator on CIFAR100. See Section 4.2 for details.

chitectures. Figure 5 shows part of the generator architectures found by NAS-caGAN-light. More architecture details are shown in Appendix C. Please refer to Appendix C.2 for comparison results with other cGAN models, and to Appendix E for the unconditional image generation experiment.

Next, we challenge our method by evaluating it on CIFAR100 which has much more classes. Thanks to the proposed mixed-architecture optimization, we can perform architecture search on CIFAR100 without additional engineering efforts compared to that on CIFAR10. Differently, we do not perform calibration or evaluate the intra-class FID scores, since there are only 500 training images for each class, which is insufficient to approximate the true image distribution.

Still, we use NAS-cGAN and NAS-caGAN to denote the class-agnostic and class-aware versions of cGAN, respectively. Table 2 summarizes the results. Again, by allowing different classes to have individual generator architectures, NAS-caGAN achieves better performance in terms of both the FID and IS scores. The searched architectures are shown in Appendix D. Though we have not achieved the state-of-the-art results, the idea of designing class-aware generators indeed brings benefits. We

Table 2: FID and IS scores on CIFAR100. [†] indicates quoted from the paper.

| Method | FID ↓ | IS ↑ |
|---|---|---|
| SN-GAN (Miyato et al., 2018) | 18.87 | 8.19 |
| *cproj* (Miyato & Koyama, 2018) | 23.20[†] | 9.04[†] |
| Multi-hinge (Kavalerov et al., 2019) | 14.62 | **13.35** |
| FQ-GAN (Zhao et al., 2020) | **8.23** | 10.62 |
| NAS-cGAN (ours) | 13.94 | 8.83 |
| NAS-caGAN (ours) | 12.28 | 9.71 |

believe that our findings can be incorporated into other methods (*e.g.*, Multi-hinge (Kavalerov et al., 2019), FQ-GAN (Zhao et al., 2020)) for better performance.

## 4.2 DIAGNOSTIC STUDIES

### 4.2.1 COORDINATION BETWEEN DISCRIMINATOR AND CLASS-AWARE GENERATORS

Based on the NAS-caGAN model, we investigate the difference between using a normal discriminator (*i.e.*, no class information) and using a class-projection (*cproj*) discriminator (Miyato & Koyama, 2018). On CIFAR10, these models report FID scores of 15.90 and 6.83, respectively, *i.e.*, the *cproj* discriminator is significantly better. We owe the performance gain to the ability that *cproj* induces more *CMConv* operators. As shown in Figure 6a, the normal discriminator leads to a sparse use of *CMConv* operators, while, as in Figure 6b, *CMConv* occupies almost half of the operators when *cproj* is used. That is, the class-aware generators should be searched under the condition that the discriminator is also class-aware. This aligns with the results obtained in (Miyato & Koyama, 2018), showing that one usually uses the *cproj* discriminator together with the generator containing conditional batch normalization operations.

### 4.2.2 WHERE SHALL WE PLACE CLASS-AWARE OPERATORS?

Last but not least, we study how our algorithm assigns the class-aware operator (*i.e.*, *CMConv*) to different positions of the searched generator architectures. Continuing the previous study, we plot the portion of *CMConv* on CIFAR100 in Figure 6c, which shows a very similar trend as the experiments on CIFAR10. In particular, the portion for the first operator (close to the input noise vector) to use *CMConv* is 100% on CIFAR10 and 85% on CIFAR100, while the portion for the last operator (close to the output generated image) is only 10% and 19% on CIFAR10 and CIFAR100, respectively. This is to suggest that class information seems very important for capturing the distribution of high-level semantics (close to input), while all classes seem to share similar low-level patterns (close to output).

To verify that the finding can indeed enhance GAN models, we experiment on BigGAN (Brock et al., 2018), a manually designed GAN model. The generator of BigGAN uses three blocks on CIFAR10, each of which contains two class-conditional operations (*i.e.*, *CBN*) by default. We try to put the *CBN* in different blocks to study the relationship between accuracy and injection position of class information. As shown in Table 3, the model using *CBN* only in the first block (index 0, close to the input noise) works better than those using *CBN* in other blocks and even the original BigGAN model (using *CBN* in all blocks). And using *CBN* only in the last block (close to output) produces inferior performance. Please refer to Appendix F for more details.

Next, we study the relationship between accuracy and the position of *CMConv* by replacing the *CBN* in BigGAN with regular *BN*, and replacing the regular convolution with *CMConv*, with other hyperparameters consistent with BigGAN. The results in Table 3 show a similar trend to the experiments of *CBN*. Besides, the best model using *CMConv* in the first block is slightly better than the best model of *CBN* (6.91 vs. 7.11).

We explain this phenomenon by the finding (Bau et al., 2018; Yang et al., 2020) that for the generator of GAN, the early and the middle layers determine the spatial layout and the category attributes, so the earlier the injection of category information, the more benefits; the final layers control the lighting and color scheme of the generated images, which are shared attributes for different classes, so using the regular convolutional layer is fine. Interestingly, our algorithm offers an alternative way to reveal this rule which was previously discovered by human experts. This indicates that our method is helpful for understanding and improving existing hand-designed generator network architectures.

Table 3: Evaluating the BigGAN architecture on CIFAR10 with *CBN* (or *CMConv*) inserted into different blocks. The table is sorted by the '*CMConv*' column. Please refer to the main texts for details.

| Block index | | | *CBN* | *CMConv* |
|---|---|---|---|---|
| 0 | 1 | 2 | | |
| ✔ | | | **7.11** | **6.91** |
| ✔ | ✔ | | 7.39 | 7.00 |
| ✔ | ✔ | ✔ | 7.55 | 7.15 |
| ✔ | | ✔ | 7.28 | 7.29 |
| | ✔ | ✔ | 8.02 | 8.06 |
| | ✔ | | 7.97 | 8.15 |
| | | ✔ | 10.60 | 9.96 |

## 5 CONCLUSIONS

In this paper, we reveal the possibility of designing class-aware generators for conditional GAN models. Though the motivation seems straightforward, non-trivial technical efforts are required to improve the performance as well as efficiency of the algorithm, in particular when the number of classes becomes large. We claim two key contributions. **First**, the search space containing regular and class-modulated convolutions eases the re-training process because all convolutional weights can be trained on the full dataset. **Second**, we design a mixed-architecture optimization mechanism so that the search and re-training processes can be performed efficiently. In a relatively small search space, the proposed Multi-Net NAS (MN-NAS) algorithm achieves FID scores of 5.85 and 12.28 on CIFAR10 and CIFAR100, respectively. Provided more computational resources, our algorithm has the potential of generating images of higher quality.

Our research leaves some open problems to the community. First, we have used the constraint that all operators are either regular or class-modulated convolution. This is to improve the efficiency of utilizing training data, but this also limits the diversity of the searched generator architectures. Second, it is interesting to jointly optimize the architectures of generator and discriminator, since we have found strong evidence of their cooperation. We leave these topics to future work.

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

---

**Algorithm 1** Searching with mixed-architecture optimization (pseudo-code in a PyTorch style).

```
 # N_op      :the number of operators in a searched edge
 # L         :the number of searched edges in an architecture
 # N_critic  :the number of times for updating the discriminator per update of the generator
 # N_policy  :update policy every N_policy iterations
 # N_c       :the number of classes

 1: for  iter, (x, y) in enumerate(data_loader):     # images x : (b, c, h, w), class label y : (b, )
        # prepare a batch of network architectures
 2:     fair_arcs = fair_arc_sampling()     # fair_arcs : (N_op, L), derived by the fair sampling
 3:     arcs = fair_arcs.repeat(b, 1)     # arcs : (b × N_op, L)

        # broadcast inputs to ensure each sample goes through fair_arcs
 4:     x = x.repeat_interleave(repeats = N_op, dim = 0)     # x : (b × N_op, c, h, w)
 5:     y = y.repeat_interleave(repeats = N_op, dim = 0)     # y : (b × N_op, )

        # update super-network parameters ω (we simplify the notation by omitting the noise z)
 6:     update_D(x, y, arcs)
 7:     if  iter % N_critic == 0:
 8:         update_G(y, arcs)

        # update policy parameters θ
 9:     if  iter % N_policy == 0:
            # sample generator architectures by the policy for all classes
10:         class_arcs = arc_sampling_by_policy()     # class_arcs : (N_c, L)
11:         reward = eval_InceptionScore(class_arcs)
12:         update_policy(reward)
     Return: policy π(a; θ)
```

---

## A  SEARCH ALGORITHM

Algorithm 1 shows the pseudo-code of the search procedure. As shown on line 2, we employ fair sampling strategy (Chu et al., 2019) to offer equal opportunity for training each child model of the super-network. However, unlike FairNAS (Chu et al., 2019) performing architecture search after the training of the super-network, our method embed the search process (policy learning) into the training loop of the super-network (shown on line 12). This is equivalent to performing a moving average for the policy parameters over the course of training. Based on NAS-caGAN model, we investigate the difference between these two search strategies. On CIFAR10, these models report FID scores of 6.83 and 7.61, respectively, *i.e.*, our proposed search strategy is better. We attribute the performance gain to the moving average for the policy parameters that could reduce the ranking noise of the weight-sharing NAS (Chen et al., 2020).

Thanks to the proposed mixed-architecture optimization, the search procedure can be very simple regardless of the number of classes. As shown on line 8, mixed-architecture optimization allows class image generation with distinct generating architectures in a single forward pass. That is to say, although each class may have varied generator architecture, these architectures can forward in parallel so that the training process is as simple as that of the original GAN (Goodfellow et al., 2014).

## B  IMPLEMENTATION DETAILS

We use the Adam optimizer (Kingma & Ba, 2014) with $\eta = 0.0001$, $\beta_1 = 0$, and $\beta_2 = 0.9$ for optimizing GAN, and $\eta = 0.00035$, $\beta_1 = 0.9$, and $\beta_2 = 0.999$ for policy learning. For the discriminator, we adopt the same architecture used in AutoGAN (Gong et al., 2019), but equip it with the *cproj* (Miyato & Koyama, 2018). The discriminator is updated five times per update of the generator. All experiments are performed on a GeForce GTX-1080Ti GPU, with the batch size set

to be 32. We search for 200K iterations and re-train for 500K iterations that are sufficient to achieve stable results.

## C  SEARCHED GENERATOR ARCHITECTURES ON CIFAR-10

### C.1  GENERATOR ARCHITECTURES FOR NAS-cGAN AND NAS-caGAN

We use *RConv*_3 × 3 and *CMConv*_3 × 3 as candidate operators. These operators and their corresponding index numbers are shown in Table 4. On CIFAR10, there are three cells for a generator architecture, each of which contains two nodes, so there are three edges to be searched in a cell and nine edges in total. The searched generator architectures of NAS-cGAN and NAS-caGAN are shown in Table 5. It can be seen that the class conditional operator (*CMConv*) tends to appear at the shallow layers (close to the input noise vector), and the class unrelated operator (*RConv*) prefers to appear at the output layers (close to the generated images). This phenomenon is definitely suggestive for future studies on the design of generator architectures.

Table 4: Candidate operators for NAS-cGAN and NAS-caGAN

| Index | Operator |
|-------|----------|
| 0 | *RConv*_3 × 3 (regular convolution) |
| 1 | *CMConv*_3 × 3 (class-modulated convolution) |

Table 5: Searched architectures for NAS-cGAN and NAS-caGAN

| Method | Class | Layer | | | | | | | | |
|--------|-------|---|---|---|---|---|---|---|---|---|
| | | 0 | 1 | 2 | 3 | 4 | 5 | 6 | 7 | 8 |
| NAS-cGAN | *All* | 1 | 1 | 0 | 1 | 0 | 0 | 0 | 0 | 0 |
| NAS-caGAN | 0 | 1 | 1 | 0 | 0 | 0 | 0 | 0 | 0 | 0 |
| | 1 | 1 | 1 | 1 | 0 | 0 | 1 | 0 | 1 | 0 |
| | 2 | 1 | 0 | 1 | 0 | 1 | 1 | 0 | 0 | 1 |
| | 3 | 1 | 1 | 1 | 0 | 0 | 1 | 1 | 1 | 0 |
| | 4 | 1 | 0 | 1 | 0 | 1 | 0 | 0 | 1 | 0 |
| | 5 | 1 | 0 | 1 | 1 | 1 | 1 | 1 | 1 | 0 |
| | 6 | 1 | 0 | 1 | 0 | 1 | 0 | 1 | 0 | 0 |
| | 7 | 1 | 0 | 1 | 1 | 1 | 0 | 0 | 1 | 0 |
| | 8 | 1 | 1 | 1 | 1 | 1 | 0 | 1 | 0 | 0 |
| | 9 | 1 | 1 | 1 | 0 | 0 | 0 | 0 | 0 | 0 |

### C.2  COMPARISON WITH STATE-OF-THE-ART cGANS

Table 6 shows some current state-of-the-art results of cGANs. We emphasize that although our results are not the best, other methods are orthogonal to ours, such as FQ-GAN (Zhao et al., 2020) and ADA (Karras et al., 2020). Combined with other approaches (*e.g.*, FQ-GAN, ADA), our method has the potential to achieve better results.

### C.3  GENERATOR ARCHITECTURES FOR NAS-caGAN-LIGHT

We add a non-parameter operator, *zero*, to the search space, and derive a light-weighted version of class-aware generators, named NAS-caGAN-light. Table 7 and 8 present the operators with index numbers and searched architectures, respectively. As shown in Table 8, the distribution of *RConv* and *CMConv* still conforms to the rules mentioned in Sec. C.1.

Class-aware generator architectures enjoy another merit that different classes could have varied computational overhead. Figure 7 shows the computational overhead of each class on CIFAR10. It can

Table 6: FID and IS scores on CIFAR10. [†] indicates quoted from the paper.

| Method | FID ↓ | IS ↑ |
|---|---|---|
| SN-GAN (Miyato et al., 2018) | 15.73 | 8.19 |
| *cproj* (Miyato & Koyama, 2018) | 17.50[†] | 8.62[†] |
| Multi-hinge (Kavalerov et al., 2019) | 6.22 | 9.55 |
| FQ-GAN (Zhao et al., 2020) | 6.54 | 9.18 |
| ADA (Karras et al., 2020) | **2.67**[†] | **10.06**[†] |
| NAS-cGAN (ours) | 6.63 | 9.00 |
| NAS-caGAN (ours) | 5.85 | 9.07 |

be seen that NAS-caGAN is superior to NAS-cGAN in terms of FID score. Both models have almost the same computational overhead for all classes because all the candidate operators are parameterized (*RConv* and *CMConv*). After incorporating the non-parameter operator, NAS-caGAN-light achieves comparable FID score with NAS-cGAN, but with less computational overhead. The visualization of the class-aware generators of NAS-caGAN-light is presented in Figure 8.

Table 7: Candidate operators for NAS-caGAN-light

| Index | Operator |
|---|---|
| 0 | *zero* |
| 1 | *RConv*_3 × 3 (regular convolution) |
| 2 | *CMConv*_3 × 3 (class-modulated convolution) |

Table 8: Searched architectures for NAS-caGAN-light

| Method | Class | Layer | | | | | | | | |
|---|---|---|---|---|---|---|---|---|---|---|
| | | 0 | 1 | 2 | 3 | 4 | 5 | 6 | 7 | 8 |
| | 0 | 2 | 0 | 0 | 1 | 1 | 0 | 1 | 2 | 1 |
| | 1 | 2 | 0 | 2 | 2 | 1 | 1 | 2 | 1 | 1 |
| | 2 | 2 | 2 | 0 | 0 | 1 | 0 | 1 | 1 | 1 |
| | 3 | 2 | 0 | 1 | 0 | 1 | 1 | 1 | 1 | 1 |
| NAS-caGAN-light | 4 | 0 | 1 | 0 | 2 | 1 | 1 | 1 | 1 | 0 |
| | 5 | 2 | 0 | 1 | 0 | 1 | 1 | 2 | 1 | 2 |
| | 6 | 2 | 2 | 0 | 0 | 1 | 0 | 1 | 2 | 1 |
| | 7 | 2 | 0 | 0 | 2 | 1 | 1 | 2 | 1 | 1 |
| | 8 | 2 | 2 | 0 | 2 | 0 | 1 | 2 | 2 | 1 |
| | 9 | 2 | 1 | 1 | 2 | 1 | 1 | 1 | 1 | 1 |

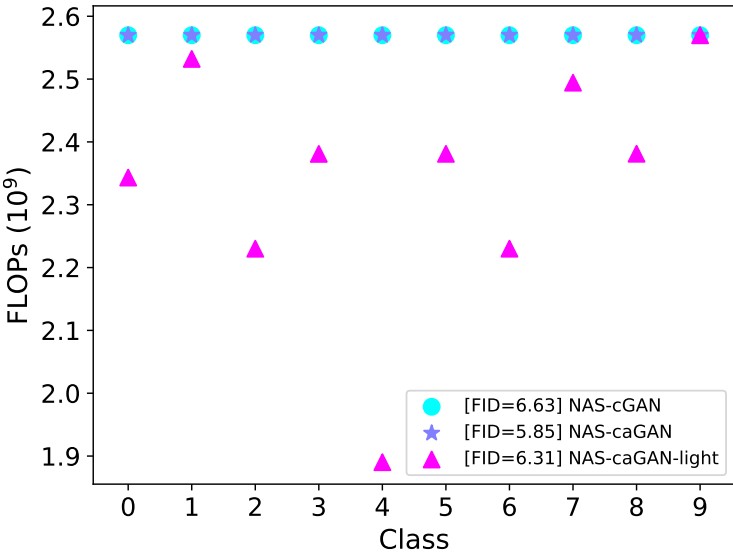

Figure 7: The computational overhead of each class on CIFAR10. Under comparable FLOPs, NAS-caGAN is superior to NAS-cGAN in terms of FID score. Under comparable FID score, NAS-caGAN-light has fewer FLOPs than NAS-cGAN for each class.

# D    SEARCHED GENERATOR ARCHITECTURES ON CIFAR-100

We perform experiments on CIFAR100, using the same search space and hyper-parameters as in CIFAR10, except increasing the number of classes to 100. That is to say, there will be more policy parameters for the search of NAS-caGAN. We emphasize that owing to the mixed-architecture optimization, the search and retraining programs can be applied to this setting with more classes without any modification. The candidate operators and searched architectures are shown in Table 9 and Table 10, respectively. One can see from Table 10 that the distribution of *RConv* and *CMConv* still conforms to the rules discussed in Sec. C.1.

Table 9: Candidate operators for NAS-cGAN and NAS-caGAN

| Index | Operator |
|---|---|
| 0 | *RConv*_3 × 3 (regular convolution) |
| 1 | *CMConv*_3 × 3 (class-modulated convolution) |

Table 10: Searched architectures for NAS-cGAN and NAS-caGAN

| Method | Class | Layer | | | | | | | | |
|---|---|---|---|---|---|---|---|---|---|---|
| | | 0 | 1 | 2 | 3 | 4 | 5 | 6 | 7 | 8 |
| NAS-cGAN | *All* | 1 | 1 | 0 | 1 | 0 | 1 | 1 | 1 | 0 |
| NAS-caGAN | 0 | 1 | 1 | 1 | 1 | 0 | 1 | 1 | 0 | 0 |
| | 1 | 1 | 0 | 0 | 0 | 0 | 1 | 0 | 1 | 1 |
| | 2 | 1 | 1 | 1 | 0 | 1 | 0 | 1 | 1 | 0 |
| | 3 | 1 | 1 | 1 | 1 | 0 | 1 | 1 | 1 | 0 |
| | 4 | 1 | 0 | 1 | 1 | 0 | 0 | 1 | 1 | 0 |
| | 5 | 1 | 1 | 1 | 0 | 0 | 0 | 0 | 0 | 0 |
| | 6 | 1 | 1 | 0 | 0 | 0 | 0 | 1 | 0 | 0 |
| | 7 | 1 | 1 | 0 | 1 | 1 | 1 | 1 | 1 | 1 |
| | 8 | 1 | 0 | 1 | 0 | 1 | 1 | 0 | 0 | 0 |

| | | | | | | | | | |
|---|---|---|---|---|---|---|---|---|---|
| 9 | 1 | 1 | 0 | 0 | 0 | 0 | 1 | 1 | 1 |
| 10 | 1 | 1 | 1 | 1 | 1 | 1 | 0 | 0 | 0 |
| 11 | 1 | 0 | 1 | 1 | 1 | 1 | 0 | 1 | 0 |
| 12 | 1 | 0 | 0 | 0 | 1 | 0 | 1 | 1 | 0 |
| 13 | 1 | 1 | 1 | 0 | 1 | 1 | 0 | 1 | 0 |
| 14 | 1 | 0 | 1 | 0 | 1 | 0 | 1 | 1 | 0 |
| 15 | 1 | 0 | 0 | 1 | 1 | 0 | 1 | 1 | 0 |
| 16 | 1 | 1 | 0 | 1 | 0 | 1 | 0 | 1 | 0 |
| 17 | 1 | 0 | 1 | 0 | 1 | 0 | 1 | 1 | 0 |
| 18 | 0 | 0 | 0 | 0 | 0 | 1 | 1 | 0 | 0 |
| 19 | 1 | 1 | 1 | 1 | 1 | 0 | 0 | 1 | 0 |
| 20 | 1 | 1 | 1 | 0 | 0 | 1 | 1 | 1 | 0 |
| 21 | 1 | 0 | 1 | 1 | 1 | 1 | 1 | 1 | 0 |
| 22 | 1 | 0 | 1 | 0 | 0 | 1 | 0 | 1 | 1 |
| 23 | 1 | 0 | 1 | 1 | 1 | 1 | 1 | 1 | 1 |
| 24 | 1 | 0 | 1 | 1 | 1 | 0 | 1 | 0 | 0 |
| 25 | 1 | 0 | 1 | 0 | 0 | 0 | 0 | 1 | 0 |
| 26 | 1 | 1 | 1 | 1 | 0 | 0 | 1 | 1 | 0 |
| 27 | 1 | 1 | 1 | 0 | 1 | 0 | 1 | 1 | 0 |
| 28 | 1 | 1 | 1 | 0 | 0 | 0 | 1 | 1 | 0 |
| 29 | 1 | 1 | 0 | 0 | 1 | 0 | 0 | 0 | 0 |
| 30 | 1 | 1 | 1 | 1 | 1 | 0 | 1 | 1 | 0 |
| 31 | 1 | 0 | 1 | 1 | 0 | 0 | 1 | 1 | 0 |
| 32 | 0 | 1 | 1 | 0 | 0 | 0 | 0 | 0 | 0 |
| 33 | 1 | 0 | 0 | 0 | 1 | 0 | 0 | 1 | 1 |
| 34 | 1 | 1 | 0 | 0 | 0 | 0 | 1 | 0 | 0 |
| 35 | 1 | 1 | 1 | 1 | 1 | 0 | 0 | 1 | 0 |
| 36 | 0 | 1 | 1 | 1 | 0 | 1 | 0 | 0 | 1 |
| 37 | 1 | 0 | 1 | 0 | 1 | 0 | 0 | 0 | 0 |
| 38 | 1 | 1 | 1 | 1 | 0 | 0 | 1 | 1 | 0 |
| 39 | 1 | 0 | 1 | 0 | 1 | 1 | 1 | 1 | 0 |
| 40 | 1 | 1 | 1 | 0 | 1 | 0 | 1 | 0 | 0 |
| 41 | 1 | 0 | 1 | 0 | 1 | 0 | 0 | 0 | 0 |
| 42 | 1 | 0 | 1 | 0 | 1 | 0 | 1 | 0 | 0 |
| 43 | 1 | 1 | 1 | 1 | 1 | 1 | 1 | 1 | 0 |
| 44 | 1 | 0 | 0 | 1 | 1 | 0 | 1 | 1 | 0 |
| 45 | 0 | 1 | 0 | 0 | 1 | 1 | 0 | 1 | 0 |
| 46 | 1 | 1 | 1 | 1 | 1 | 0 | 0 | 0 | 0 |
| 47 | 0 | 0 | 1 | 0 | 0 | 0 | 0 | 1 | 0 |
| 48 | 1 | 1 | 1 | 0 | 1 | 1 | 0 | 1 | 0 |
| 49 | 1 | 1 | 0 | 1 | 1 | 1 | 1 | 1 | 1 |
| 50 | 1 | 1 | 1 | 1 | 1 | 0 | 1 | 1 | 0 |
| 51 | 1 | 0 | 0 | 0 | 1 | 1 | 1 | 1 | 0 |
| 52 | 1 | 1 | 0 | 0 | 0 | 0 | 1 | 1 | 0 |
| 53 | 1 | 1 | 1 | 1 | 1 | 1 | 1 | 0 | 1 |
| 54 | 1 | 0 | 1 | 0 | 0 | 0 | 0 | 1 | 0 |
| 55 | 1 | 0 | 0 | 0 | 1 | 1 | 0 | 0 | 0 |
| 56 | 1 | 1 | 1 | 0 | 0 | 0 | 0 | 0 | 0 |
| 57 | 1 | 1 | 1 | 0 | 0 | 1 | 0 | 0 | 0 |
| 58 | 1 | 1 | 0 | 0 | 0 | 0 | 0 | 1 | 0 |
| 59 | 1 | 1 | 1 | 0 | 0 | 0 | 1 | 1 | 0 |
| 60 | 1 | 1 | 1 | 0 | 1 | 0 | 1 | 1 | 0 |
| 61 | 1 | 0 | 1 | 0 | 1 | 0 | 0 | 0 | 0 |
| 62 | 0 | 0 | 1 | 0 | 0 | 1 | 0 | 0 | 1 |
| 63 | 0 | 0 | 0 | 1 | 0 | 0 | 1 | 1 | 0 |
| 64 | 1 | 0 | 1 | 1 | 1 | 1 | 1 | 0 | 1 |
| 65 | 0 | 0 | 1 | 1 | 0 | 0 | 1 | 0 | 0 |
| 66 | 1 | 0 | 1 | 0 | 0 | 1 | 1 | 1 | 0 |

| | | | | | | | | | |
|---|---|---|---|---|---|---|---|---|---|
| 67 | 1 | 0 | 1 | 1 | 0 | 1 | 1 | 1 | 1 |
| 68 | 0 | 1 | 1 | 0 | 0 | 0 | 1 | 0 | 0 |
| 69 | 1 | 1 | 1 | 0 | 1 | 0 | 1 | 1 | 0 |
| 70 | 1 | 0 | 1 | 1 | 0 | 0 | 0 | 0 | 1 |
| 71 | 1 | 1 | 1 | 1 | 1 | 1 | 1 | 1 | 0 |
| 72 | 1 | 0 | 0 | 1 | 0 | 1 | 1 | 1 | 1 |
| 73 | 1 | 1 | 0 | 1 | 0 | 1 | 1 | 1 | 1 |
| 74 | 0 | 1 | 1 | 1 | 0 | 0 | 1 | 1 | 0 |
| 75 | 1 | 0 | 1 | 1 | 0 | 1 | 1 | 1 | 0 |
| 76 | 1 | 0 | 1 | 0 | 1 | 1 | 0 | 1 | 0 |
| 77 | 1 | 0 | 0 | 1 | 1 | 0 | 1 | 0 | 0 |
| 78 | 1 | 0 | 1 | 0 | 1 | 0 | 0 | 1 | 0 |
| 79 | 1 | 1 | 1 | 0 | 1 | 1 | 0 | 0 | 0 |
| 80 | 1 | 1 | 1 | 0 | 0 | 0 | 1 | 0 | 0 |
| 81 | 1 | 1 | 0 | 0 | 1 | 0 | 0 | 0 | 0 |
| 82 | 0 | 1 | 1 | 0 | 0 | 1 | 1 | 1 | 0 |
| 83 | 1 | 0 | 1 | 0 | 0 | 1 | 0 | 0 | 0 |
| 84 | 1 | 0 | 0 | 0 | 0 | 0 | 0 | 0 | 0 |
| 85 | 1 | 1 | 0 | 0 | 1 | 0 | 0 | 1 | 0 |
| 86 | 1 | 0 | 0 | 0 | 1 | 0 | 0 | 1 | 0 |
| 87 | 1 | 1 | 1 | 0 | 0 | 1 | 0 | 0 | 1 |
| 88 | 1 | 0 | 1 | 0 | 0 | 0 | 1 | 1 | 0 |
| 89 | 1 | 1 | 1 | 0 | 1 | 0 | 0 | 1 | 0 |
| 90 | 1 | 1 | 0 | 0 | 1 | 0 | 0 | 1 | 0 |
| 91 | 0 | 1 | 1 | 0 | 1 | 1 | 1 | 0 | 1 |
| 92 | 0 | 1 | 1 | 0 | 1 | 1 | 1 | 1 | 1 |
| 93 | 1 | 1 | 1 | 0 | 0 | 1 | 0 | 0 | 1 |
| 94 | 1 | 0 | 1 | 0 | 0 | 0 | 0 | 1 | 0 |
| 95 | 1 | 1 | 0 | 1 | 1 | 1 | 1 | 1 | 0 |
| 96 | 0 | 1 | 0 | 0 | 1 | 0 | 0 | 0 | 0 |
| 97 | 1 | 1 | 1 | 0 | 1 | 0 | 1 | 0 | 0 |
| 98 | 1 | 0 | 1 | 1 | 1 | 1 | 0 | 0 | 0 |
| 99 | 0 | 1 | 1 | 0 | 0 | 1 | 1 | 0 | 0 |

## E   UNCONDITIONAL IMAGE GENERATION ON CIFAR10

We study the effectiveness of the search space and the search algorithm. We design two class-agnostic versions based on the architecture used in class-aware setting. The first one is to directly set all operators to be *RConv*, and the second is to search a common architecture for all classes, with each the operator on each edge chosen among *RConv*, *skip-connect*, and *zero*. Note that we remove the *CMConv* operator to guarantee that class information is NOT used at all.

Table 11: Comparison between existing automatically designed GAN methods and our algorithm in the setting that all classes share the same generator architecture. No class label is used at all.

| Method | Params (M) | FID $\downarrow$ | IS $\uparrow$ |
|---|---|---|---|
| AGAN (Wang & Huan, 2019) | 20.1 | 30.50 | $8.29 \pm 0.09$ |
| AutoGAN (Gong et al., 2019) | 4.4 | 12.42 | $8.55 \pm 0.10$ |
| AdversarialNAS (Gao et al., 2019) | 8.8 | 10.87 | $\mathbf{8.74} \pm 0.07$ |
| baseline (full) | 6.0 | 12.26 | $8.61 \pm 0.07$ |
| NAS-GAN (searched) | 5.4 | $\mathbf{10.80}$ | $8.32 \pm 0.09$ |

The discriminator is also class-agnostic with the same architecture used in AutoGAN (without *cproj*).

Results are summarized in Table 11. We deliver two messages. First, our search space that contains 9 selectable operators seems small, but has sufficient ability to find powerful models and compete against recently published methods. Second, the searched NAS-GAN preserves most parameterized operator (*RConv*), showing that GAN needs a sufficient amount parameters to achieve good performance. This supports the design of our search space, *i.e.*, the competition is held between two parameterized operators. The searched architecture is shown in Table 13.

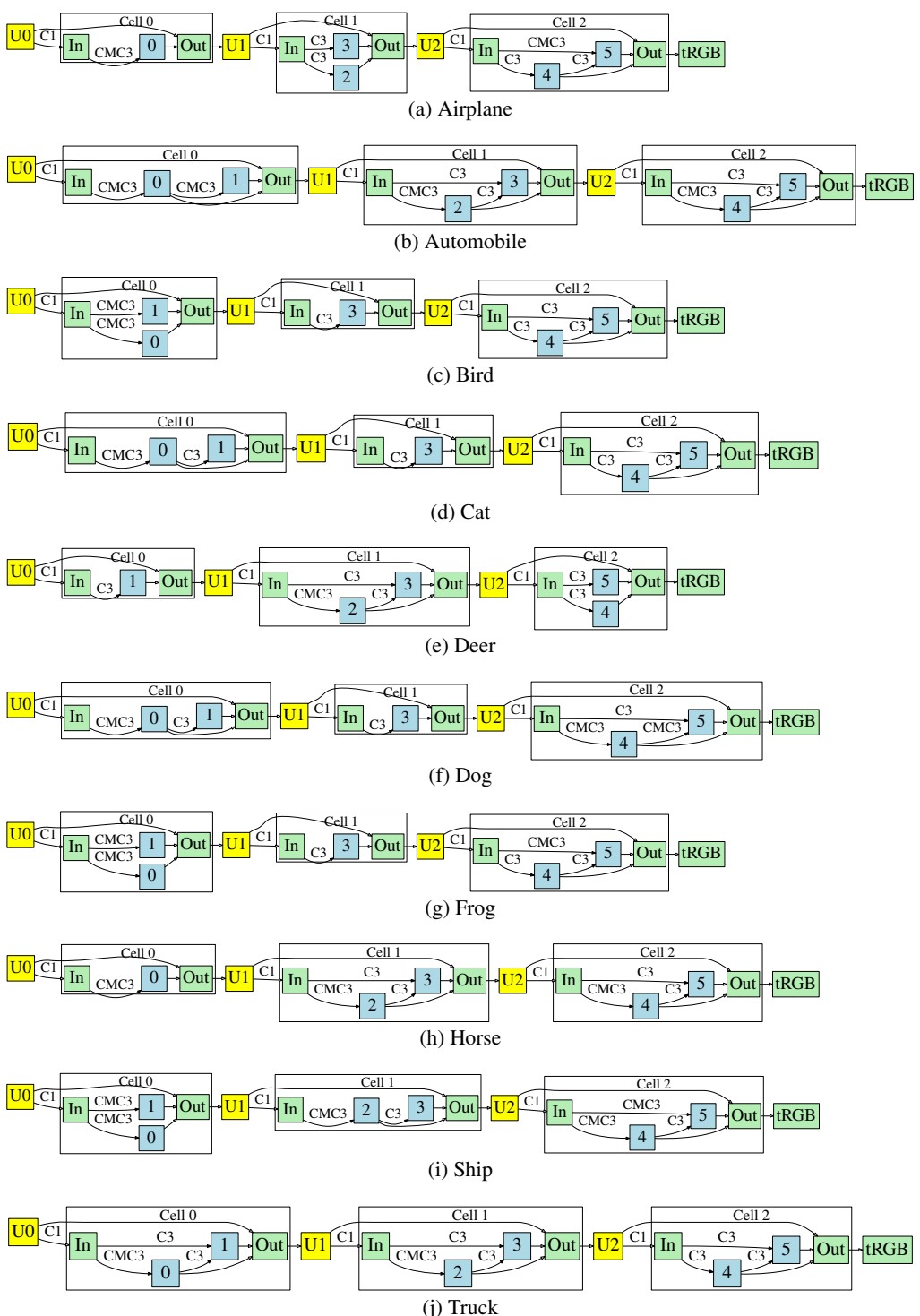

Figure 8: Visualization of class-aware generators of NAS-caGAN-light

Table 12: Candidate operators for NAS-GAN

| Index | Operator |
| --- | --- |
| 0 | *zero* |
| 1 | *skip-connect* |
| 2 | *RConv_3 × 3* (regular convolution) |

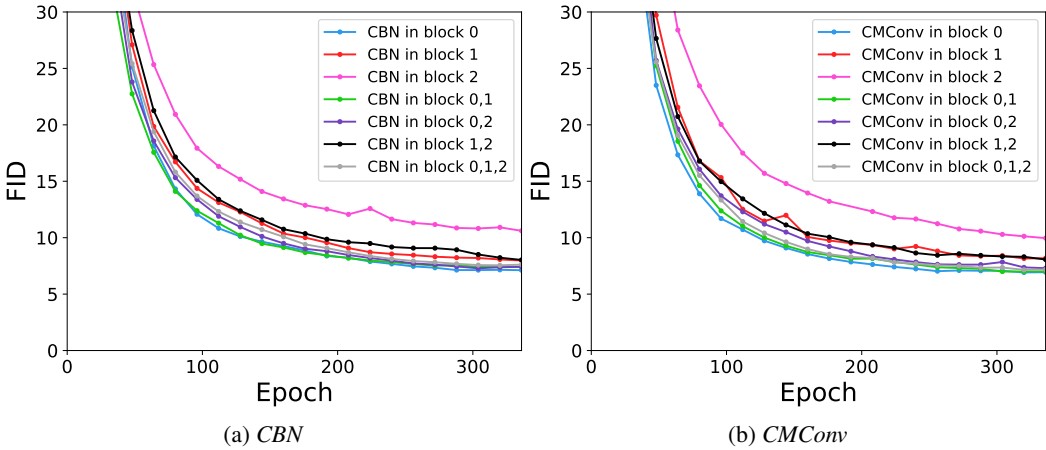

(a) *CBN*  (b) *CMConv*

Figure 9: Evaluating the effect of the inserted position of *CBN* (or *CMConv*) in BigGAN on CI-FAR10. Inserting *CBN* (*CMConv*) in different positions does affect accuracy. This phenomenon is consistent with the rule we summarize through our Multi-Net NAS method.

Table 13: Architectures for baseline and NAS-GAN. The candidate operators and their corresponding indices are shown in Table 12

| Method | Class | Layer | | | | | | | | |
|---|---|---|---|---|---|---|---|---|---|---|
| | | 0 | 1 | 2 | 3 | 4 | 5 | 6 | 7 | 8 |
| baseline (full) | *All* | 2 | 2 | 2 | 2 | 2 | 2 | 2 | 2 | 2 |
| NAS-GAN (searched) | *All* | 2 | 2 | 1 | 2 | 2 | 2 | 2 | 2 | 2 |

# F  IMPACT OF INJECTION POSITION OF CLASS INFORMATION ON ACCURACY

We use BigGAN, a manually designed GAN model, to study the impact of injection position of class information on accuracy. On CIFAR10, the generator of BigGAN has three blocks, each of which contains two class-conditional operations (*i.e.*, *CBN*). We do controlled experiments by changing the inserted position of *CBN* (blocks that do not use *CBN* use regular *BN* as a substitute). On the other hand, we replace the regular convolution in the blocks with *CMConv*, replace all *CBN* with regular *BN*, and study the effect of the inserted position of *CMConv* on the accuracy. As shown in Figure 9, The injection position of class information does have an impact on accuracy. This phenomenon has not been paid attention by previous work. We discover this phenomenon through our Multi-Net NAS.

# G  RESULTS OF CLASS CONDITIONAL IMAGE GENERATION

We show some generated images of NAS-caGAN models trained on CIFAR10 and CIFAR100, in Figure 10 and Figure 11, respectively. In the figures, each row corresponds to samples of one class. All the samples are randomly sampled rather than cherry-picked.

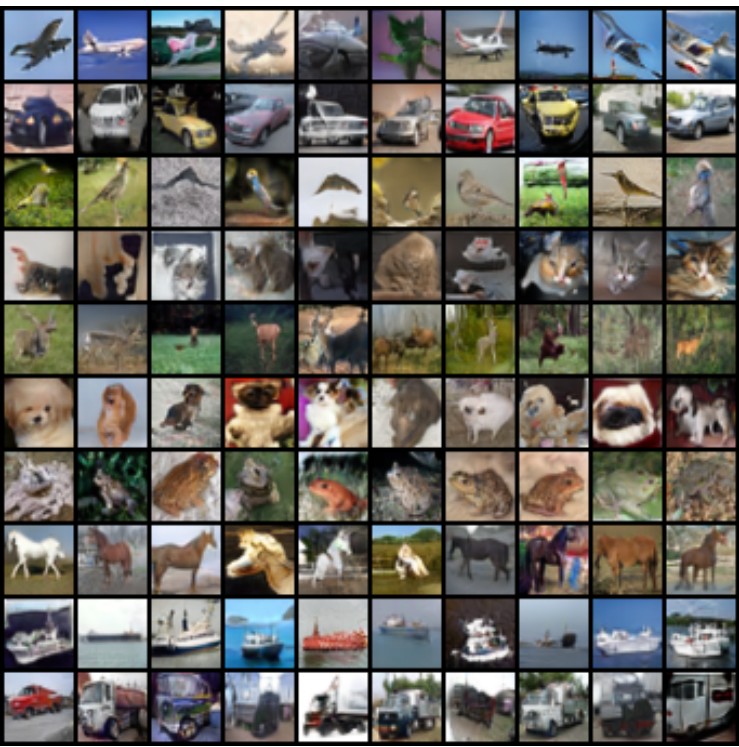

Figure 10: Generated images of NAS-caGAN model trained on CIFAR10. Each row corresponds to samples of the same class.

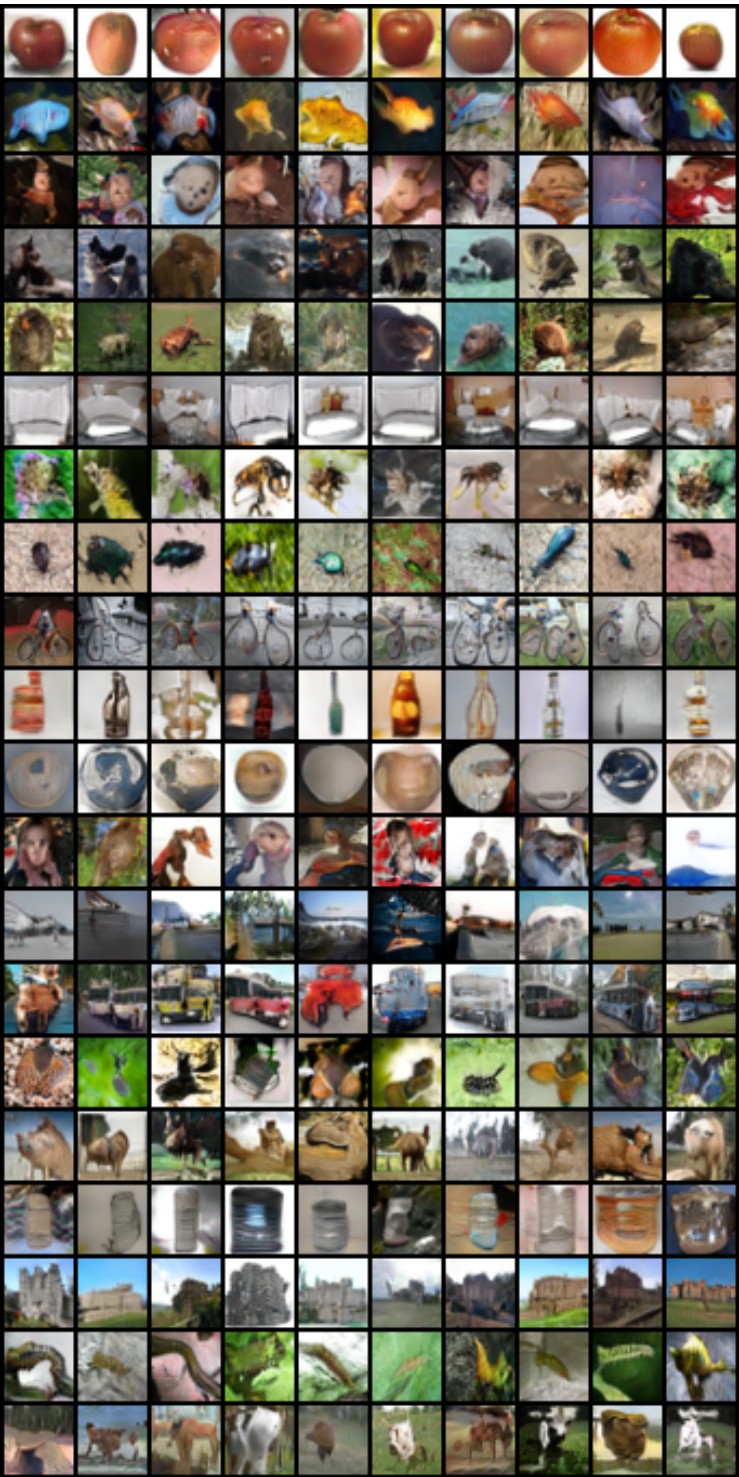

Figure 11: Generated images of NAS-caGAN model trained on CIFAR100. Each row corresponds to samples of the same class. Due to space limitations, we only show samples of 20 classes without cherry-picking.

