# OpenReview forum: "Searching towards Class-Aware Generators for Conditional Generative Adversarial Networks"
_ICLR.cc/2021/Conference — Reject_

### Official Review · AnonReviewer1 · 2020-10-19
**This paper applies Neural Architecture Search to Conditional GAN. The Class-Modulated Conv is proposed to condition the search space.**

**Rating:** 5
**Confidence:** 4

**Review:**

Pros:
This paper designs class-aware generators (increasing flexibility) for cGAN by RL-based neural architecture search (NAS) algorithm. GAN models with better flexibility are promising to yield better performances.

Class-Modulated convolution (CMconv) is designed to increase model flexibility while enabling data sharing among classes, increasing the efficiency of training data.

Mixed-architecture optimization is presented to ease the training procedure of multiple class-aware generators.

Cons:

The proposed Class-Modulated Conv also inserts the class embedding to a common convolution, which is similar to a regular BN. The architectures for different categories are still weight-sharing, which is quite a common approach in NAS. Thus, this work may be treated as applying NAS to cGAN.

For searching network for GANs, the main challenge lies in how to provide stable and efficient supervision as a reward. Note that during training, the generator (G) and discriminator (D) play as rivals. Searching for the architectures of G and choosing IS as a reward only helps the G compete against D. I wonder whether it is the optimal choice, so the authors need to consider the settings in [1], i.e., updating the architectures for both G and D.

Many NAS algorithms for GAN models are relevant to this work, but none of them is evaluated against the proposed method in the experiments. I think the authors should add competing results from AdversarialNAS [1] and AutoGAN. Specifically, the Intra FIDs on CIFAR10, FID, and IS scores on CIFAR100 of the two methods should be reported.

Neither time complexity nor space complexity is reported. Considering that the resolution of outputs is only 32$\times$32, I wonder whether the proposed method is prohibitively expensive to be applied to practical applications that require higher resolution. Note that the resolution for most generation tasks is at least 256$\times$256 or 128$\times$128. I suggest the authors report the training and testing times, as well as the consumed GPU resources.


[1] Gao, Chen, Yunpeng Chen, Si Liu, Zhenxiong Tan, and Shuicheng Yan. "Adversarialnas: Adversarial neural architecture search for gans." CVPR, pp. 5680-5689. 2020.

[2] Gong, Xinyu, Shiyu Chang, Yifan Jiang, and Zhangyang Wang. "Autogan: Neural architecture search for generative adversarial networks." ICCV, pp. 3224-3234. 2019.

---

> ### Author Response · Authors · 2020-11-24
> **Response to AnonReviewer1**
>
> Thank you for your comments. We provide our summary of the paper above. Please refer to it for common questions. Below are the responses to the specific comments
>
> **The authors need to consider the settings in AdversarialNAS, i.e., updating the architectures for both G and D.**
>
> Although we believe that this would be outside the scope of our paper because our core contribution is not the NAS method, we are still willing to discuss this in detail. In fact, the current RL-based search algorithms for GAN are limited to only search the generator network architecture but not the discriminator network architecture. The reason is that for the current weight-sharing RL-based NAS method, it is difficult for us to get a metric to evaluate the quality of the discriminator during the searching process. For a sampled generator architecture, we can obtain its IS or FID, but for a sampled discriminator, we cannot get any valuable metrics. However, AdversarialNAS can search for the discriminator and generator at the same time because it is based on a differentiable search method.
>
> **The authors should add competing results from AdversarialNAS and AutoGAN. Specifically, the Intra FIDs on CIFAR10, FID, and IS scores on CIFAR100 of the two methods should be reported.**
>
> We respectfully disagree with the reviewer. AdversarialNAS and AutoGAN are unconditional GANs. Let them be compared with NAS-caGAN (a conditional GAN) in terms of intra FID is not fair. In fact, we have provided the comparison results of NAS-GAN (an unconditional GAN model searched by our method) with AdversarialNAS and AutoGAN in the appendix (Table 11).
>
> **Training and testing times, as well as the consumed GPU resources.**
>
> The search time is within two days using one GTX 1080ti GPU. The retraining time is about three days using one GPU (GTX 1080ti).

---

### Official Review · AnonReviewer2 · 2020-10-26
**Overall a good work, but the motivation is not quite convincing, and the novelty is somewhat limited. Experiments on higher-resolution datasets are expected.**

**Rating:** 5
**Confidence:** 4

**Review:**

This paper proposes an interesting method that adopts NAS to search multiple class-aware generator architectures for cGAN instead of class-agnostic type. A search space containing both normal and class-modulated convolutions are introduced to simplify the process of re-training. Besides, this paper design a mixed-architecture optimization to specifically address the computational burden issue under the setting of a multi-net search. The search results also give some insights about constructing cGAN models.


Strengths:
- The perspective of adopting the NAS method to explore the class-aware generator architectures for conditional GANs is relatively novel and interesting, although there are some related works about searching unconditional GANs.
-The proposed flexibility and safety search space is effective to address the categories grow issue.
- The developed mixed-architecture optimization is a clever way to improve the efficiency of the search and re-training process.
- The authors conduct extensive experiments and give some interesting insight/discussion about the results.

Weaknesses:
- There are no innovative approaches toward neural architecture search are proposed, and this work only focuses on how to bring existing RL-based NAS methods to cGANs while overcoming some issues.
- The motivation for using distinct architecture to generate images for each class instead of using one architecture is unclear. Most of the existing cGANs in noise-to-image and image-to-image translation settings employ CBN or AdaIN to embed conditional information to a unified generator.
- The idea is similar to the dynamic routing/inference networks such as Blockdrop [1], and it needs a related discussion about the difference.
- The experiments are only conducted on low-resolution datasets. In my view, the GANs search algorithm needs to be verified on high-resolution datasets, instead of still continuing to achieve marginal performance improvements on small datasets.

Wu, Zuxuan, et al. "Blockdrop: Dynamic inference paths in residual networks." Proceedings of the IEEE Conference on Computer Vision and Pattern Recognition. 2018.

---

> ### Author Response · Authors · 2020-11-24
> **Response to AnonReviewer2**
>
> Thank you for the constructive comments. We provide our summary of the paper above. Please refer to it for common questions. Below are the responses to the specific comments
>
> **There are no innovative approaches toward neural architecture search are proposed.**
>
> Yes, we did not propose new methods for NAS. The only difference for the NAS method is that each class has its own controller parameters, so that the generator network architecture of all classes can be obtained with only one search. However, we emphasize that NAS is only a tool to realize the idea of class-aware generators, not our key contribution.
>
> **The motivation for using distinct architecture to generate images for each class instead of using one architecture is unclear.**
>
> We summarize our main contributions in the above summary. We hope that the reviewer will read it. To sum up, the purpose of class-aware generators is not to boost performance but to help us understand cGANs more deeply. This is the first work to realize the idea of class-aware generators, with non-trivial efforts. Besides, by analyzing the searched class-aware generators, we have obtained some interesting findings. We sincerely hope that reviewers appreciate this work.
>
> **The idea is similar to the dynamic routing/inference networks such as Blockdrop [1], and it needs a related discussion about the difference.**
>
> To be honest, I had not read the paper while I did this work. We thank the reviewer for the insightful suggestions, and we are actively willing to continue to update our paper.
>
> **The GANs search algorithm needs to be verified on high-resolution datasets, instead of still continuing to achieve marginal performance improvements on small datasets.**
>
> I agree with you very much. But for this paper, since our goal is not to use NAS to boost the performance of cGANs, the search algorithm is not the core contribution of the paper. We explain this in the summary above in detail.

---

### Official Review · AnonReviewer3 · 2020-10-27
**An interesting idea but may lack of some supportive experiments (see cons).**

**Rating:** 5
**Confidence:** 4

**Review:**

##########################################################################

Summary:

This paper proposes a framework NAS-caGAN that adopts RL-based NAS to search the optimal class-aware generator architecture by directly optimizing the Inception Score (IS) using the  REINFORCE algorithm, and leverages the mixed-architecture optimization to mitigate the training data sparsity of each category. The authors design a Class-Modulated Convolution to allow for the weight-sharing among different searched architectures. The proposed NAS-caGAN outperforms the model that employs searched class-agnostic architecture on CIFAR 10 and achieves better results compared with cproj (Miyato & Koyama, 2018) on CIFAR 100.

##########################################################################

Reasons for score:

Overall, I vote for rejecting, but I am happy to modify the score if the authors could provide further details about my concerns. My major concerns are about the baseline choice, computational costs, and the evidence to support the method’s utility (see cons below).  Hopefully the authors can address my concern in the rebuttal period.

##########################################################################

Pros:
1. The idea of using an RL-based NAS to search for the optimal architecture of different categories is quite interesting and empirically demonstrates its superior effect when compared with the searched identical generator architecture regardless of the category class.
2. Overall, the paper is well written and technically sound.
3. Good ablation test that highlights the utility of the class-projection (cproj) discriminator over a standard GAN’s discriminator.
4. Good illustration of the proportion of different operators (i.e., RConv and CMConv) in each layer of the class-aware architecture.

##########################################################################

Cons:
1. The key concern about the paper is the lack of experiments to validate the utility of the proposed method compared with the previous work. Despite the paper asserting that the usefulness of searching for distinct architecture for different categories, the paper only compares the **searched** class-agnostic model (NAS-cGAN) with the proposed class-aware one (NAS-caGAN), and ignores the comparison with the literature on CIFAR 10, such as (Brock et al., 2018; Kavalerov et al., 2019; Zhao et al., 2020) .
2.  Another concern is the baseline presented.
(1) Most recent works such as (Kavalerov et al., 2019; Zhao et al., 2020) selected BigGAN (Brock et al., 2018) as their baseline. Considering the limited results (not achieving the state-of-the-art on neither of the presented metrics), a deeper analysis and performance comparison of the proposed framework would have been helpful to argue for its effectiveness.
(2) It is interesting to inject CMConv operation at the first layer of BigGAN, which achieves a better result than other operator settings. Why not use BigGAN as the baseline?
3. Could the authors provide the details about computational and **time** costs of the proposed method? How much time/resources would this method take to search?
4. In the Sec.4.1, the paper claims that this method could benefit other works. Considering the lack of convincing experiments to support this argument, it is doubtful about its correctness.
5. For the “moving average”, it would be better to provide more details about it. The paper claims the usage of it for training stability, but never mentions the details about it, which seems not very clear to me.
6. In the Sec.2, it would have been nice to supplement the details compared with the recent literature of GANs with NAS.

#########################################################################

Minor comments:
1. Table 3:  The reported evaluation score is not mentioned, which might be FID.
2. The “moving average” has not been well addressed in the main paper.
3. Section 3.1: it would be easier to follow if the authors could paraphrase the last two paragraphs.

---

> ### Author Response · Authors · 2020-11-24
> **Response to AnonReviewer3**
>
> Thank you for the comments, but we cannot fully agree with the comments. We provide our summary of the paper above. Please refer to it for common questions. Below are the responses to the specific comments
>
>
>
>
> **The key concern is lack of comparison with the literature such as Brock et al., 2018; Kavalerov et al., 2019; Zhao et al., 2020, ...not use BigGAN as the baseline.**
>
> We respectfully disagree with the reviewer. We can compare with them but we think it is unnecessary because our main contribution is not to achieve state of the art for cGANs. We recommend the reviewer to read the summary we provide above. **Please do not reject this paper on the pretext of performance.**
>
> **Details about computational and time costs of the proposed method.**
>
> The search time is within two GPU days using one GTX 1080ti GPU.
>
> **In the Sec.4.1, the paper claims that this method could benefit other works. ...doubtful about its correctness.**
>
> The reviewer might have overlooked our key contribution. Please refer to the summary we provide above.
>
> **For the “moving average”, it would be better to provide more details about it.**
>
> The reviewer might have overlooked some results. In fact, the detail of "moving average" for the policy parameters was provided in Appendix A in the original submit.
>
> **In the Sec.2, it would have been nice to supplement the details compared with the recent literature of GANs with NAS.**
>
> Thank you for your comment, but we cannot do that because the space in the original manuscript is too limited.
>
> **Table 3: The reported evaluation score is not mentioned, which might be FID.**
>
> Yes, the score is FID. We thank the reviewer for pointing out this issue.

---

### Official Review · AnonReviewer5 · 2020-10-28

**Rating:** 5
**Confidence:** 3

**Review:**

Summary:
This paper proposes to use neural architecture search (NAS) to automatically discover useful conditional generative adversarial network (cGAN) architectures. Specifically, this work aims to find a dedicated architecture for each class.

Strengths:
-Paper is well written.
-Demonstrates that optimizing architectures for each class yields some improvement over using a single architecture for all classes.
-Architectures learned by the NAS reveal insights about how to use existing building blocks, such as where best to place feature modulation layers in the network.

Weaknesses:
-No random baseline (see [1]).
-Given how close NAS-cGAN and NAS-caGAN are in terms of performance, confidence intervals should be reported to confirm that improvement is significant.
-Majority of improvement comes from fine-tuning on each class individually. It is unclear how much of this improvement is simply due to additional capacity.
-Complexity of the model appears to be disproportionate to the improvement in performance (lots of implementation effort for a somewhat small gain in performance).

Recommendation and Justification:
While I think this paper is well written, after reading it I am not convinced of the usefulness of the core idea, which is that generator architectures should be class-aware. It is never explained why it might be desirable for each class to have a distinct generator network, and I cannot think of any reason why this may be the case aside from increased model capacity. For this reason I think that this paper is currently marginally below the acceptance threshold, but look forward to the author's explanation.

Clarifying Questions:
-How is calibration performed exactly? Is this procedure the same for NAS-cGAN and NAS-caGAN? Is a separate copy of weights fine-tuned for each class?  If a new copy of weights needs to be created for each class, proper comparison would be to a non-NAS model with up to n_classes times more model capacity than the base model.

[1] Li, Liam, and Ameet Talwalkar. "Random search and reproducibility for neural architecture search." Uncertainty in Artificial Intelligence. PMLR, 2020.

---

> ### Author Response · Authors · 2020-11-24
> **Response to AnonReviewer5**
>
> Thank you for the helpful comments. We provide our summary of the paper above. Please refer to it for common questions. Below are the responses to the specific comments
>
> **About the usefulness of the core idea, namely searching class-aware generators for cGANs**
>
> I understand the reviewer's concerns. In fact, we also think that the performance gain of NAS-caGAN is limited. However, we emphasize that the key contribution of this paper is not to improve performance of cGANs. Searching class-aware generators can help us discover statistical rules, so that we can better understand cGAN models. This is the key contribution of this paper. For more details, please refer to the summary above we provide to everybody.
>
> **How is calibration performed exactly? Is this procedure the same for NAS-cGAN and NAS-caGAN? Is a separate copy of weights fine-tuned for each class?**
>
> Each class will have a separate copy of weights at the beginning of the fine-tining. After fine-tuning, the weights of each class will be different. This procedure is applied to both NAS-cGAN and NAS-caGAN. It can only show that NAS-caGAN gets more benefits from calibration compared to NAS-cGAN.

---

> > ### Comment · AnonReviewer5 · 2020-11-24
> > **Response to Authors**
> >
> > Thank you for the nice summary! This is very helpful.
> >
> > However, I do not think this claim is accurate:
> > > "For example, for image-to-image tasks, SPADE [2] used an unconditional GAN."
> >
> > As far as I am aware, the architecture used in the SPADE paper is indeed conditional, since the semantic mask used for conditioning is concatenated with the real/generated image before input to the discriminator (see Figure 15 in that paper).
> >
> > By the definition outlined in the original cGAN paper [1], a GAN is conditional if both the generator and discriminator have access to some extra information. It is impossible to have a conditional GAN with an unconditional discriminator. Therefore, I am wary of the value of the first claimed statistical rule, since it has been known for as long as cGANs have been around.
> >
> > [1] Mirza, Mehdi, and Simon Osindero. "Conditional generative adversarial nets." arXiv preprint arXiv:1411.1784 (2014).
> > [2]T. Park, M.-Y. Liu, T.-C. Wang, and J.-Y. Zhu, “Semantic Image Synthesis with Spatially-Adaptive Normalization”

---

> > > ### Author Response · Authors · 2020-11-25
> > > **Response to the comment**
> > >
> > > **"SPADE [2] used an unconditional GAN"**
> > >
> > > Thank you for your reply.
> > >
> > > What I mean is that SPADE does not explicitly use a classification-based discriminator. For example, the SPADE‘s discriminator only judges whether the input image is real or fake. Besides, I think simply concatenating semantic mask with input cannot make the generator and discriminator of cGANs fully cooperate. A possible improvement scheme, I think, is that we can design class embedding (namely, learnable parameters) for the generator, and then use a classification-based loss for the discriminator such as Multi-hinge loss [1]. In this way, the discriminator and generator will fully cooperate.
> > > Looking forward to your reply！
> > >
> > > [1]I. Kavalerov, W. Czaja, and R. Chellappa, “cGANs with Multi-Hinge Loss” Available: http://arxiv.org/abs/1912.04216.

---

### Official Review · AnonReviewer4 · 2020-10-30
**Class-awareness mechanism involved in the NAS approach**

**Rating:** 5
**Confidence:** 3

**Review:**

This paper proposes an interesting idea that adopts NAS to find a distinct architecture for each class based on cGAN framework. Within the framework, the paper also proposes an operator, Class-Modulated convolution (CMconv), to allow the training data to be shared among different architectures, so as to balance the training data across classes. The proposed method leverages a Markov Decision Process (MDP) in the search algorithm, and learns the sampling policy for NAS. Comprehensive experiments demonstrate the class-aware NAS can outperform class-agnostic NAS.

- The paper is well written with sufficient figures and plots.
- The proposed idea is straightforward and convincing.
- Rich experiments and analysis are conducted. Implementation details are clearly described in the appendix.
- I like the figures architecture searched specifically for different classes.

However, I still have some concerns:

- From my point of view, the proposed CMconv has exactly the same architecture as the one in Karras et al., instead of using class embedding as input. Please clarify the difference and clearly point out in the paper.

- Quantitative results in Table 2 seem not promising. The proposed method is not compatible with the SOTA. Although the paper claims that the proposed idea can be applied to the existing methods and performs better, there’s no evidence showing that.

- It’d be better to also list infra-FIDs of one existing unconditioned GAN method in the Table 1, for the lower bound / baseline of the experiments.

- For fair comparison, it’s better to have a table with quantitative results of IS/FID on CIFAR-10, listing the existing methods with conditioned/unconditioned, such as AutoGAN, style GAN etc.

Overall, the proposed method is interesting, NAS by MDP with class-aware functionality, which can ideally outperform the class-agnostic based method. The experiments are comprehensive, with strong ablation study and analysis. However, it still requires some clarification and convincing experiments to demonstrate the performance.

I'm willing to rate higher if the concerns are addressed.

---

> ### Author Response · Authors · 2020-11-24
> **Response to AnonReviewer4**
>
> Thank you for the helpful comments and suggestions for improvements.  We thank reviewer for appreciating the class-aware generator architectures. We like these architectures too. We provide our summary of the paper above. Please refer to it for common questions. Below are the responses to the specific comments
>
> **Clarify the difference between the CMConv and the modulated convolution in Karras et al [1].**
>
> The only difference between CMConv and modulated convolution [1] is that CMConv uses learnable class embeddings as the input of modulated convolution, so that class information can be injected into the convolution kernel parameters. We will further clarify and emphasize this point in our paper.
>
> **Quantitative results in Table 2 seem not promising**
>
> Table 2 shows some results of cGANs on CIFAR100. NAS-caGAN is inferior than FQ-GAN and Multi-hinge GAN. We think the reason may be that NAS-caGAN did not utilize feature quantization [2] and multi-hinge loss [3] for the discriminator, which are effective to improve FID and IS respectively. However, we want to emphasize that the main contribution of this paper is not to achieve good performance on CIFAR, but that we use NAS to search for a distinct generator network architecture for each class. By analyzing all the class-aware generator architectures, we found some interesting rules for conditional GANs. To realize this idea, we have overcome many difficulties, including introducing CMConv for sharing data with ordinary convolution, multi-net NAS obtaining multiple network architectures through one search, and mixed-architecture optimization enabling training multiple class-aware generators easier. We sincerely hope that reviewers will not reject this paper just for reasons of performance.
>
> **List infra-FIDs of one existing unconditioned GAN method in the Table 1**
>
> We answer the reason why we did not compare with unconditional GANs in Table 1. If we list the performance of unconditional GAN in Table 1, we are sure that some reviewers will criticize us for comparing conditional GAN with unconditional GAN. In order not to cause controversy, we think it is better not to list unconditional GANs in Table 1.
>
> **Have a table with quantitative results of IS/FID on CIFAR-10**
>
> The reviewer might have overlooked some results. In fact, we have included two tables in the appendix (please refer to Table 6 and Table 11).
>
>
> [1]T. Karras, S. Laine, M. Aittala, J. Hellsten, J. Lehtinen, and T. Aila, “Analyzing and Improving the Image Quality of StyleGAN”
>
> [2]Y. Zhao, C. Li, P. Yu, J. Gao, and C. Chen, “Feature Quantization Improves GAN Training”
>
> [3]I. Kavalerov, W. Czaja, and R. Chellappa, “cGANs with Multi-Hinge Loss”

---

### Author Response · Authors · 2020-11-23
**Summary of this paper from authors.**

We sincerely thank all reviewers, ACs, and PCs for their time and efforts. We summarize the paper here and highlight findings that may be beneficial to the community.Then we clarify questions from every individual review.

**What did we do?**

We design class-aware generators for conditional GANs. Although the idea seems straightforward, non-trivial technical efforts are required. We list these technologies as follows.

* We use NAS to automatically design the generator network architecture for each class. In particular, we propose a multi-net NAS algorithm, which can obtain the generator network architectures of all classes through only one search.

* Inspired by Karras et al. [1], we introduce a new operation named class-modulated convolution (CMConv). This operation allows the training data to be shared among different generator architectures and thus alleviates the issue of limited training data for each class.

* We develop mixed-architecture optimization, such that multiple class-aware generators can be optimized at the same time. This allows us to easily apply the idea of class-aware generators to datasets with more classes such as CIFAR100 in the paper.

**The core idea of this paper is to make the generator architecture be class-aware. Is it useful?**

Yes, we are sure that the class-aware idea is valuable. However, the value is not to boost performance, but to help us discover some key rules about cGANs. We emphasize two rules discovered by analyzing the class-aware generators. If we do not use class-aware generators, it is difficult for us to clearly discover these statistical rules.

1. Coordination between the discriminator and the generator is important for cGANs. In particular, the unconditional discriminator leads to sparse use of conditional operators in the generator, while there will be more conditional operators in the generator when using the conditional discriminator (i.e., projection discriminator employed in the paper). This phenomenon was discovered by analyzing the searched class-aware generators. We emphasize that this discovery is beneficial to the community, and it can help us improve some of the existing GAN-based methods. For example, for image-to-image tasks, SPADE [2] used an unconditional GAN. If we want to improve the performance of SPADE, we first need to design a class conditional operator for the generator, and then let the discriminator also have an ability to distinguish categories. We think this is a direction worth trying because our paper has shown that the coordination between generators and discriminators is critical.

2. The position of injecting class information into the generator is important for cGANs. Specifically, we find that the conditional operator is more likely to appear in the early stage (close to the input noise) of the generator. This rule is also discovered by analyzing the class-aware generators and is beneficial to the community. For example, when doing style transfer tasks, we must be careful in which layers to inject content information and in which layers to inject style information. Perhaps we are not the first to discover this phenomenon, but we verified this phenomenon by means of NAS and class-aware generators.

**Weaknesses**

* We acknowledge that the performance of the current NAS-caGAN on CIFAR100 is ordinary compared to the current mainstream cGAN models (refer to Table 2 in the paper). However, the main purpose of class-aware is not to achieve state of the art, but to enable us to better understand the properties of cGANs by means of analyzing class-aware generators, so that we know what to focus on when using cGANs.

* There are no high-resolution experiments. The experiment of using GANs to generate high-resolution images requires a lot of computing resources. Besides, it will become more complicated when combined with NAS. Therefore, currently we do not conduct high-resolution experiments. However, we believe that this does not affect the main contribution of this paper.

All in all, this paper has implemented a new idea, namely using class-aware generators for cGANs. Although the experiments are limited to CIFAR10 and CIFAR100, we found some interesting rules through analyzing the searched class-aware generators. This is the first work to realize the idea of class-aware generators, with non-trivial efforts. **Please do not reject this paper on the pretext of not achieving state of the art.** We believe that readers will get inspiration for cGANs from this paper. Finally, if you find the above comments to be useful and employ any of them in the future, please cite this paper. Thanks a lot!


[1]T. Karras, S. Laine, M. Aittala, J. Hellsten, J. Lehtinen, and T. Aila, “Analyzing and Improving the Image Quality of StyleGAN”

[2]T. Park, M.-Y. Liu, T.-C. Wang, and J.-Y. Zhu, “Semantic Image Synthesis with Spatially-Adaptive Normalization”

---

### Decision · Program_Chairs · 2021-01-07
**Final Decision**

**Decision:**

Reject

**Comment:**

This paper investigates the use of class-conditional architectures in GANs. It achieves this by employing neural architecture search (NAS) on top of reinforcement learning. Their main contribution is a “flexible and safe” search space; experiments are carried out on CIFAR-10 and -100. Standard performance results are augmented by diagnostic studies.

This paper received a total of five reviews which, remarkably, yielded the same assessment: the paper had merits but was marginally below the acceptance threshold. In general, the reviewers thought the idea was interesting and straightforward, experiments extensive and the paper was clearly written. The primary concerns brought up were novelty (i.e. just cGAN + NAS?, R1 and R2), minimal performance gain (R4, R5), unclear motivation (R2), lack of comparisons --- including to other NAS for GAN methods --- (R1,R3), limited to low-resolution datasets (R2,R3), no reporting of time or space complexity (R1,R3), unclear where improvement comes from --- no control for capacity --- (R5).

On the point about comparing to NAS + GAN works, the authors responded, stating that the NAS + GAN methods brought up by the reviewer were unconditional GAN methods and pointed out that they made unconditional GAN comparisons in the Appendix.

The authors also emphasized to multiple reviewers that the point of the paper was not to improve NAS. Interestingly, they also made a comment to R5 that the point of the paper was not to improve performance of cGANs, but to improve understanding of them.

The reviewers are unanimous in that this paper falls just below the bar for the reasons outlined above. Following the discussion phase, I see no reason to overturn their recommendation. I hope that the authors can use the feedback from these reviews to improve this paper and re-submit it.